# Characteristics and expression profiles of circRNAs during abdominal adipose tissue development in Chinese Gushi chickens

Wenjiao Jin[1]⊙, Yinli Zhao[2]⊙, Bin Zhai[1]⊙, Yuanfang Li[1], Shengxin Fan[1], Pengtao Yuan[1], Guirong Sun[1], Ruirui Jiang[1], Yanbin Wang[1], Xiaojun Liu[1], Yadong Tian[1], Xiangtao Kang[1], Guoxi Li[1] *

1 College of Animal Science and Technology, Henan Agricultural University, Zhengzhou, Henan Province, P. R. China, 2 College of Biological Engineering, Henan University of Technology, Zhengzhou, Henan Province, P.R. China

⊙ These authors contributed equally to this work.
* liguoxi0914@126.com

**Data Availability Statement:** The raw data from this study have been stored in NCBI Sequence Read Archive (accession numbers PRJNA528858 and PRJNA551368).

## Abstract

Circular RNAs (circRNAs) play important roles in adipogenesis. However, studies on circRNA expression profiles associated with the development of abdominal adipose tissue are lacking in chickens. In this study, 12 cDNA libraries were constructed from the abdominal adipose tissue of Chinese domestic Gushi chickens at 6, 14, 22, and 30 weeks. A total of 1,766 circRNAs were identified by Illumina HiSeq 2500 sequencing. These circRNAs were primarily distributed on chr1 through chr10 and sex chromosomes, and 84.95% of the circRNAs were from gene exons. Bioinformatic analysis showed that each circRNA has 35 miRNA binding sites on average, and 62.71% have internal ribosome entry site (IRES) elements. Meanwhile, these circRNAs were primarily concentrated in TPM < 0.1 and TPM > 60, and their numbers accounted for 18.90% and 80.51%, respectively, exhibiting specific expression patterns in chicken abdominal adipose tissue. In addition, 275 differentially expressed (DE) circRNAs were identified by comparison analysis. Functional enrichment analysis showed that the parental genes of DE circRNAs were primarily involved in biological processes and pathways related to lipid metabolism, such as regulation of fat cell differentiation, fatty acid homeostasis, and triglyceride homeostasis, as well as fatty acid biosynthesis, fatty acid metabolism, and glycerolipid metabolism. Furthermore, ceRNA regulatory networks related to abdominal adipose development were constructed. The results of this study indicated that circRNAs can regulate lipid metabolism, adipocyte proliferation and differentiation, and cell junctions during abdominal adipose tissue development in chickens through complex ceRNA networks between circRNAs, miRNAs, genes, and pathways. The results of this study may help to expand the number of known circRNAs in abdominal adipose tissue and provide a valuable resource for further research on the function of circRNAs in chicken abdominal adipose tissue.

**Funding:** This study was funded by a grant from the National Natural Science Foundation of China (32072692 and 31572356), the Program for Innovation Research Team of Ministry of Education (IRT16R23) and the Scientific Studio of Zhongyuan Scholars (No. 30601985).

**Competing interests:** The authors declare that they have no competing interests.

## Introduction

Chickens are an economically important agricultural animal [1]. Excessive fat accumulation in the abdominal cavity not only reduces feed utilization efficiency but also affects meat quality in poultry production [2]. Therefore, demonstrating the molecular mechanism governing chicken abdominal adipose tissue development is of great significance to chicken genetic breeding and production. CircRNAs are a type of nonphosphorylated, single-stranded, covalently closed noncoding RNA that is widely present in various tissues and cell types of different species [3]. Many studies have shown that circRNAs may be involved in the regulation of various biological processes and affect cell physiology through various molecular mechanisms [4]. In particular, some studies have shown that circRNAs are also involved in the regulation of adipogenesis. For example, circFUT10 has been reported to promote adipocyte proliferation and inhibit adipocyte differentiation through sponge adsorption of let-7 [5]. Therefore, circRNAs can be employed as new targets to elucidate the molecular mechanism underlying chicken abdominal fat development.

CircRNAs are produced by a special reverse splicing process that covalently closes the 5' end of the upstream exon and the 3' end of the downstream exon of pre-mRNA, which primarily exist in the cytoplasm [6]. It is known that more than 20% of the expressed genes in the genome can produce circRNAs, which present notable abundance, stability, and diversity of expression profiles in various types of tissues and cells [3]. CircRNAs can be used as miRNA- or RNA-binding protein decoys to regulate gene expression or protein translation [4] and thus play pivotal roles in various physiological activities and diseases. Therefore, circRNAs have recently become a heavily researched topic in the field of RNA. At present, research on circRNAs related to fat deposition is primarily focused on identifying novel circRNAs from adipose tissue or adipocytes in different anatomic sites. For example, Sun (2020) detected the circRNA expression profiles of preadipocytes and adipocytes in human visceral adipose tissue by microarray technology. Compared with preadipocytes, 2,215 circRNAs were significantly upregulated, and 1,865 were downregulated, in adipocytes [7]. In the Chinese Erhualian pig, Liu (2019) identified many DE circRNAs in adipose tissue of intramuscular, subcutaneous, peritoneal and mesenteric adipose tissues [8] and between subcutaneous preadipocytes and adipocytes [9]. In yaks (*Bos grunniens*), a previous study found 211 DE circRNAs from adipose tissue in yaks of two age groups [10], and another study identified 7,203 circRNAs and 136 DE circRNAs during yak adipocyte differentiation [11]. In addition, Huang (2019) identified 5,141 circRNAs and 252 DE circRNAs from the adipose tissue of adult and young Chinese buffaloes [12]. These studies indicated that many circRNAs were involved in regulating adipose tissue development or adipose deposition in animals. Although these results helped to elucidate the molecular mechanisms underlying adipose tissue development or fat deposition, current research has primarily concentrated on humans and a few other species. In addition, only a few studies have described the mechanism of specific circRNAs in adipogenesis [5, 13, 14]. At present, the function of circRNAs in the development or deposition of animal adipose tissue has not been fully elucidated. The expression of circRNAs is known to have obvious tissue or cell specificity [15]. Therefore, the identification of circRNAs related to adipose tissue development in economically important animals is an important step to better study circRNA function in the formation of adipose tissue-related traits.

In view of the importance of chickens in poultry production, the roles played circRNAs in the formation of economic traits have been studied increasingly frequently by researchers. Previous studies have identified many novel circRNAs. For instance, Shen (2019, 2020) quantified circRNAs from theca cells [16] and granulosa cells [17] during follicular development in chickens, counting 14,502 and 1,1642, respectively. Zhang (2019) studied the differential expression

profiles of circRNAs in chicken hepatocytes in GHR antisense transcript overexpression groups and control groups and identified 4,772 circRNAs and 92 DE circRNAs [18]. Ouyang (2018) identified 13,377 circRNAs and 462 DE circRNAs from the leg muscles of female Xinghua chickens of different embryo ages and at one day post-hatching [19]. In addition, several studies have demonstrated the mechanism of action of specific circRNAs in chicken skeletal muscle satellite cells or myoblasts [20–23]. These studies have proven that many circRNAs participate in the formation and regulation of economically important traits in chickens, which helps to elucidate the molecular mechanism governing the formation and regulation of economically important traits in chicken. To the best of our knowledge, however, there have been no reports characterizing circRNAs in relation to adipose tissue development in chickens.

Gushi chickens are native Chinese chickens that originate from Gushi County, Henan Province, China. These chickens have been bred on a large scale due to their excellent meat quality. To elucidate the role played by circRNAs in the development of abdominal fat, we constructed the circRNA transcriptome profile of 6-, 14-, 22- and 30-week-old Gushi chickens and the regulatory network of circRNAs related to abdominal fat development in Gushi chickens. The results of this study may not only help to elucidate the posttranscriptional regulation of abdominal fat deposition in chickens but also provide basic data for resource protection and the mining of useful characteristics in Gushi chickens.

## Materials and methods

### Ethics statement

All animal experiments were performed according to the program approved by the Institutional Animal Care and Use Committee (IACUC) of Henan Agricultural University (Permit Number: 17–0118).

### Experimental animals and total RNA extraction

In this study, domestic Gushi chickens were used as experimental animals. The feeding method was as described by Fu (2018) [24]. Before reaching 14 weeks of age, chickens were fed a diet containing 18.5% crude protein and 12.35 MJ/kg energy, and after reaching the age of 14 weeks, they were fed a diet containing 15.6% crude protein and 12.75 MJ/kg energy. At 6, 14, 22, and 30 weeks of age, three individuals were randomly selected at each developmental stage. After dissection, abdominal adipose tissues were collected and stored at -80˚C. Total RNA was extracted by using TRIzol reagent (Takara, Dalian, China) according to the instructions of the manufacturer. A 1% agarose gel electrophoresis and Agilent Bioanalyzer 2100 system (Agilent, Santa Clara, CA, USA) were used to evaluate the quality of the total RNA.

### Library construction and sequence analysis

The initial total RNA content of each sample used for library construction was 5 μg. rRNA was removed from total RNA by an Epicentre Ribo-Zero™ kit (Illumina, San Diego, USA). A short fragment of 250–300 bp was prepared as a template for the synthesis of double-stranded cDNA. Subsequently, the A tail was added, the sequencing adapter was connected, and a chain-specific cDNA library was obtained by PCR amplification and enrichment. The library was diluted to 1 ng/μl, and an Agilent 2100 (Agilent, Santa Clara, CA, USA) was used to detect the insert size of the library to meet the expected size. Quantitative PCR (qPCR) was used to quantify the library to ensure that the effective concentration of the library was > 2 nm. Finally, the qualified library was sequenced on the Illumina HiSeq 2500 platform (Novogene, Beijing, China). The quality of sequencing data was evaluated rigorously. The adaptor

sequences and low-quality sequences in raw data were eliminated. The Q20, Q30 and GC contents of clean data were calculated. The obtained high-quality reads were used for subsequent analysis. The paired-end clean reads were compared with the reference genome by Bowtie software (Releases 1.2.3) [25].

## Identification of circRNAs

CircRNAs were identified using find_circ [26] and CIRI2 [27] software. Circos software (v0.69–9) [28] was used to construct circos diagrams. The reverse shear method was used to extract the connection points that were not aligned with reads. The prediction analysis of circRNAs for each software was performed according to the system default parameters. Only those sequences containing more than 2 independent junction-spanning reads that met the Breathnach-Chambon rule (GU/AG rule) were identified as potential circRNAs. To reduce false positives, the shared circRNAs in the results predicted by the two software programs were identified as circRNAs based on their position in the chromosomes.

## Identification of DE circRNAs

The read count of each circRNA in different libraries was calculated according to back-spliced reads per million mapped reads, and the transcripts per million clean tags (TPM) were used to correct it. The normalized expression value was calculated by the following formula: normalized expression level = (read count × 1,000,000)/libsize (libsize is the sum of circRNA read count). Based on the read count, DE circRNAs were analyzed by DESeq2 software (v3.12) [29] in six comparison groups, including W14 *vs*. W6, W22 *vs*. W6, W30 *vs*. W6, W14 *vs*. W22, W14 *vs*. W30, and W22 *vs*. W30. $p < 0.05$ was considered to be the threshold criterion for determining the DE circRNAs.

## Potential function analysis of circRNAs

First, a functional enrichment analysis was performed on the parental genes of DE circRNAs. The GoSeq R package (v3.12) [30] was used for enrichment analysis of Gene Ontology (GO), and Kyoto Encyclopedia of Genes and Genomes (KEGG) enrichment analysis was conducted using KOBAS software (v3.0) [31]. $p < 0.05$ was used as the threshold for evaluating GO terms or KEGG pathways for significant enrichment. Second, miRNA target sites were predicted by miRanda software (Releases 3.3a), and the analysis of the interaction between circRNAs and miRNAs was completed. In addition, IRESfinder software (v1.1.0) [32] was utilized to predict and identify the internal ribosome entry site (IRES) on the circRNA sequences to determine the circRNAs with the potential to translate polypeptides or proteins.

## Interactive network analysis of circRNA-miRNA-gene pairs

The mRNA and miRNA transcriptome profiles of Gushi chicken abdominal adipose tissue at 6, 14, 22, and 30 weeks were also constructed using the same tissue samples. Based on these sequencing data, miRanda software (Releases 3.3a) was used to predict the binding sites of DE miRNAs on circRNA and miRNA target genes. By Pearson's correlation analysis, 5,455 DE circRNA-miRNA-gene pairs were identified according to the expression levels of circRNAs, miRNAs and genes in abdominal adipose tissue of Gushi chickens. The 431 target genes in these circRNA-miRNA-gene pairs were annotated with GO and KEGG using DAVID online tools (https://david.ncifcrf.gov/). In addition, the circRNA-miRNA-gene pairs were selected from the pathways associated with lipid metabolism, cell proliferation and differentiation, and cell junctions, and the interaction networks were constructed by Cytoscape software (v3.2).

## Reliability verification of circRNAs

First, reverse transcription PCR analysis and Sanger sequencing were used to verify the circRNA sequences from RNA-seq analysis. Total RNA was extracted from abdominal adipose tissue samples of Gushi chickens at different weeks. After the RNA quality was qualified, it was mixed in equal amounts for cDNA synthesis. Ten circRNAs, namely, gga_circ_0000348, gga_circ_0003969, gga_circ_0005065, gga_circ_0001623, gga_circ_0000833, gga_circ_0004577, gga_circ_0003686, gga_circ_0003828, gga_circ_0003244 and gga_circ_0002520, were randomly selected and verified by PCR amplification using divergent primers (S1 Table). Subsequently, the PCR product fragment size was detected by gel electrophoresis; the sequence was determined by Sanger sequencing; finally, DNAMAN software (v6.25) was used to compare the sequence obtained by Sanger sequencing with RNA-seq data and the chicken reference genome.

Second, the resistance of the above nine circRNAs to RNase R (Geneseed, Guangzhou, China) digestion was detected by quantitative real-time PCR (qRT-PCR). Before cDNA synthesis, total RNA was treated with RNase R (RNR-07250, Epicentre). Subsequently, using an RT-PCR kit (Takara, Dalian, China), RNA treated with RNase R and RNA without RNase R treatment were separately synthesized into cDNA. All qRT-PCR analyses were completed in accordance with the SYBR Green kit (Takara, Dalian, China) product instructions. All reactions were repeated three times. The $2^{-\Delta\Delta\mathrm{ct}}$ method [33] was used to calculate the relative expression of circRNAs, and the linear gene glyceraldehyde-3-phosphate dehydrogenase (*GAPDH*) was used as a control.

## Plasmid construction and dual luciferase reporter assay

The miR-215-5p mimics and mimics NC were synthesized by GenePharma (Shanghai, China). For the construction of the dual-luciferase reporter vector, wild-type and mutated sequences in the 3′UTR of *NCOA3* and the partial region of gga_circ_0002520 (named WT-NCOA3, Mut-NCOA3, gga_circ_2520-WT and gga_circ_2520-Mut, respectively), which contain the binding sites of miR-215-5p, were synthesized and inserted into psiCHECK2 vectors (Promega, Madison, WI, USA) according to instructions using XhoI and NotI restriction sites.

After planting DF-1 cells evenly in 96-well plates and growing to 70%, four groups (wild-type psiCHECK2 plasmids and miR-215-5p mimics as the treatment, mutated psiCHECK2 plasmids and miR-215-5p mimics, wild-type psiCHECK2 plasmids and miR-215-5p mimics NC, mutated type psiCHECK2 plasmids and miR-215-5p mimics NC) were set and cotransfected. To confirm the target relationship between gga_circ_2520 and miR-215-5p, another dual-luciferase reporter assay method was employed. Three comparison groups (WT-NCOA3; WT-NCOA3 and miR-215-5p mimics; WT-NCOA3, miR-215-5p mimics and gga_circ_2520-WT) were set and cotransfected. After 48 h, a dual luciferase assay system kit (Promega, Madison, WI, USA) and fluorescence/multidetection microplate reader (BioTek, Winooski, VT, USA) were employed to detect luminescent signals. Firefly luciferase activities were normalized to Renilla luminescence in each well. Detailed protocols were described in the manufacturer's instructions.

## Results and discussion

### Overview of library sequencing

A total of 12 cDNA libraries were constructed using abdominal fat tissue obtained from Gushi chickens aged 6, 14, 22, and 30 weeks. Raw reads obtained from each library ranged from 93,729,158 to 116,176,278. After removing adapter reads and low-quality reads, the amount of

**Table 1. Sequencing results and quality evaluation in 12 libraries.**

| Library | Raw_reads | Clean_reads | Clean_bases | Error_rate(%) | Q20(%) | Q30(%) | GC_content(%) |
|---------|-----------|-------------|-------------|---------------|--------|--------|---------------|
| W6_1 | 93,729,158 | 107,398,162 | 13.49G | 0.02 | 97.31 | 93.15 | 47.88 |
| W6_2 | 105,587,052 | 95,285,904 | 15.21G | 0.01 | 97.40 | 93.34 | 48.53 |
| W6_3 | 105,361,016 | 103,378,214 | 15.14G | 0.01 | 97.43 | 93.38 | 50.05 |
| W14_1 | 111,668,572 | 107,385,996 | 16.11G | 0.02 | 96.69 | 91.48 | 48.72 |
| W14_2 | 99,103,024 | 111,513,774 | 14.29G | 0.02 | 96.62 | 91.38 | 46.24 |
| W14_3 | 107,542,426 | 93,216,400 | 15.51G | 0.02 | 96.56 | 91.18 | 49.01 |
| W22_1 | 111,838,654 | 110,402,524 | 16.11G | 0.01 | 97.40 | 93.32 | 51.52 |
| W22_2 | 116,176,278 | 105,675,972 | 16.73G | 0.01 | 97.45 | 93.41 | 51.56 |
| W22_3 | 97,137,262 | 113,365,746 | 13.98G | 0.02 | 96.29 | 90.64 | 47.91 |
| W30_1 | 113,865,916 | 89,940,330 | 16.56G | 0.02 | 97.17 | 92.78 | 48.89 |
| W30_2 | 108,976,258 | 101,379,478 | 15.85G | 0.02 | 97.17 | 92.78 | 47.93 |
| W30_3 | 116,860,112 | 100,927,314 | 17.00G | 0.02 | 97.17 | 92.80 | 47.62 |

**Note:** The circRNA libraries of abdominal adipose samples from 6-, 14-, 22-, and 30-week-old Gushi chickens are represented by W6, W14, W22, and W30, respectively. There were three biological repeats for each developmental stage. The same abbreviations are used in the other tables and figures.

clean base data obtained by each library ranged from 13.49 G to 16.73 G, accounting for more than 95%. In addition, reference genome mapping of the sequence was also completed. The results showed that the percent range of total mapped reads/clean reads in different libraries was 78.27–82.8%, of which the percent range of multiple mapped reads/clean reads was 2.21–8.29%, and the percent range of uniquely mapped reads/clean reads was 74.02–80.08%. More-over, 64.46–70.7% reads were aligned to the protein-coding region, and 19.01–26.09% of the reads were compared to the intron region. The sequence output and quality evaluation of each library are shown in Table 1 and S1 Fig.

## CircRNAs expressed in chicken abdominal adipose tissues

A total of 1,766 novel circRNAs were identified from the 12 cDNA libraries in Gushi chicken abdominal adipose tissue (S1 File). In total, 1,637, 1,711, 1,735 and 1,663 circRNAs were expressed in the abdominal adipose tissue of Gushi chicken at 6, 14, 22, and 30 weeks of age, respectively. These circRNAs were primarily located in exons of genes with an average ratio of 84.95% followed by intergenic regions (9.62%) and introns (5.43%). As shown in S2 Fig, the full length of these circRNAs ranged from 185 nt to 99,591 nt, and their splicing length was 43–1842 nt. The full length of 543 circRNAs was more than 10,000 nt, and the length range of 1,071 circRNAs was 1,000–9,999 nt. Moreover, these circRNAs were distributed on chr1 through chr28 and sex chromosomes, especially in chr1 (19.03%), chr2 (14.04%), chr3 (13.93%), chr4 (7.19%) chr5 (5.83%), chrZ (4.70%), chr6 (4.13%), chr7 (3.96%), chr8 (3.40%), chr10 (2.66%), and chr9 (1.98%) (S2 Fig).

Previously, several studies have reported circRNAs expressed in adipose tissue, preadipo-cytes and adipocytes in different anatomical parts of humans and several other species, such as pigs, yaks, and buffalo [7–12]. These studies indicate that a large quantity of circRNAs partici-pate in the regulation of fat tissue development or adipogenesis. The results of this study are consistent with the findings of previous studies, suggesting that the development or deposition of chicken abdominal fat is also regulated by many circRNAs. Ten circRNAs were randomly selected and verified by reverse transcription PCR analysis, Sanger sequencing and RNase R digestion tests. These results proved that the identification of circRNAs in this study was accu-rate and reliable (Figs 1A, 1B, 9A and 9B). Therefore, the results of this study provide a basis

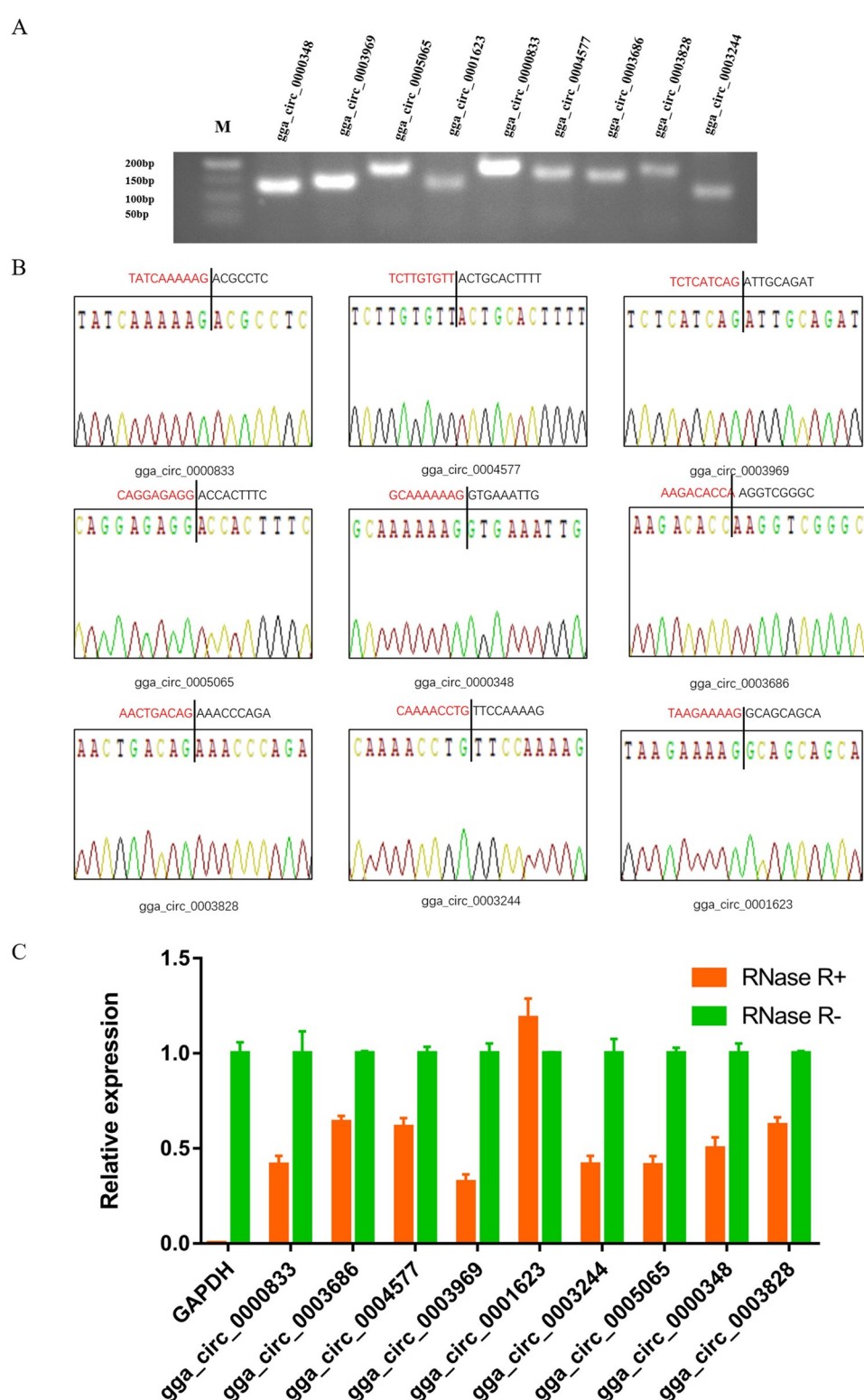

**Fig 1. Verification of 9 randomly selected circRNAs.** (A) Electrophoretogram of the PCR products. M: marker 500. (B) The head-to-tail junctions of 9 circRNAs were confirmed by Sanger sequencing. The black line indicates the position of reverse splicing. The area marked with red (or black) is the end (or start) sequence of the circRNAs (junction). (C) qRT-PCR was used to detect the resistance of circRNA to RNase R digestion. *GAPDH* was used as a linearity control. Data are expressed as means ± SEM (n = 3).

for further research on the molecular mechanism for the regulation of chicken abdominal adipogenesis by circRNAs.

## Expression profiles of circRNAs during abdominal adipose development

The expression of circRNAs is known to be tissue- and developmental stage-specific [4, 34]. The TPM value distribution of circRNA in this study showed that the circRNA expression levels of all samples had two peaks, indicating that the expression patterns of circRNAs in the four stages of Gushi chicken abdominal fat development were highly consistent (Fig 2A). These circRNAs were mainly concentrated in TPM < 0.1 and TPM > 60, and their numbers accounted for 18.90% and 80.51%, respectively. This finding indicated that circRNAs had specific expression patterns in chicken abdominal adipose tissue. This specific expression pattern is similar to that of circRNAs in porcine intramuscular, subcutaneous, retroperitoneal, and mesenteric adipose tissues [8].

However, the expression profiles of circRNAs were different in different developmental stages of abdominal adipose tissue in Gushi chickens. In particular, there were many circRNAs with notable changes in expression in each developmental stage (Fig 2B). Meanwhile, we found that 1,518 of the identified circRNAs were expressed in all four developmental stages. The expression profiles of these circRNAs exhibited clear temporal characteristics and were divided into eight different subclusters (Fig 2C and S3 Fig). We also found that one, three, six, and two circRNAs were expressed specifically in abdominal adipose tissue at 6, 14, 22, and 30 weeks of age, respectively (S1 File). In addition, 275 DE circRNAs were identified from six combinations (S2 File). Accordingly, the DE circRNAs in W14 *vs.* W6, W22 *vs.* W6, W30 *vs.* W6, W22 *vs.* W14, W30 *vs.* W14, W30 *vs.* W22 combinations were 50, 80, 87, 61, 67, 57, respectively. These DE circRNAs were clustered together repeatedly within the group, indicating that the differences within the group were less than the differences between the groups and proving that the data were reliable (Fig 2D). The expression patterns of these DE circRNAs were divided into four different modes and had clear temporal characteristics at different developmental stages of abdominal fat tissues (Fig 2E). The distribution of DE circRNAs showed that there was no common DE circRNA between all six comparison combinations or any five comparison combinations (Fig 3). Only gga_circ_0001526 and gga_circ_0002515 were DE in four comparison combinations, including W22 *vs.* W6, W30 *vs.* W6, W22 *vs.* W14, and W30 *vs.* W14. These findings indicated that the expression profile of circRNAs in chicken abdominal adipose tissue had clear developmental stage specificity.

## Potential function of circRNAs in chicken abdominal adipose tissues

A large number of circRNAs are derived from protein-coding genes [35–37]. In this study, we observed that 1,579 circRNAs were derived from 1,076 known genes, and the parental genes of some circRNAs were closely related to lipid metabolism or adipogenesis. For example, the host gene *ASXL2* of gga_circ_0005102 can increase the activity of liver X receptor alpha and play a key role in lipid metabolism [38, 39], and the host gene cytochrome P450 family 39 subfamily A member 1 (*CYP39A1*) for gga_circ_0005166 can participate in lipoprotein transport and cholesterol metabolism [40]. Moreover, the diacylglycerol O-acyltransferase 2 (*DGAT2*) gene, which can encode three circRNAs, gga_circ_0002513, gga_circ_0002515 and gga_-circ_0002517, plays a vital role in the synthesis of triacylglycerol [41]. To further elucidate the potential roles played by circRNAs in chicken abdominal fat deposition, functional enrichment analysis of the parental genes of DE circRNAs was performed. The results showed that the functions of DE circRNAs were significantly different in different developmental stages of abdominal adipose tissues. The biological functions of DE circRNAs in abdominal adipose

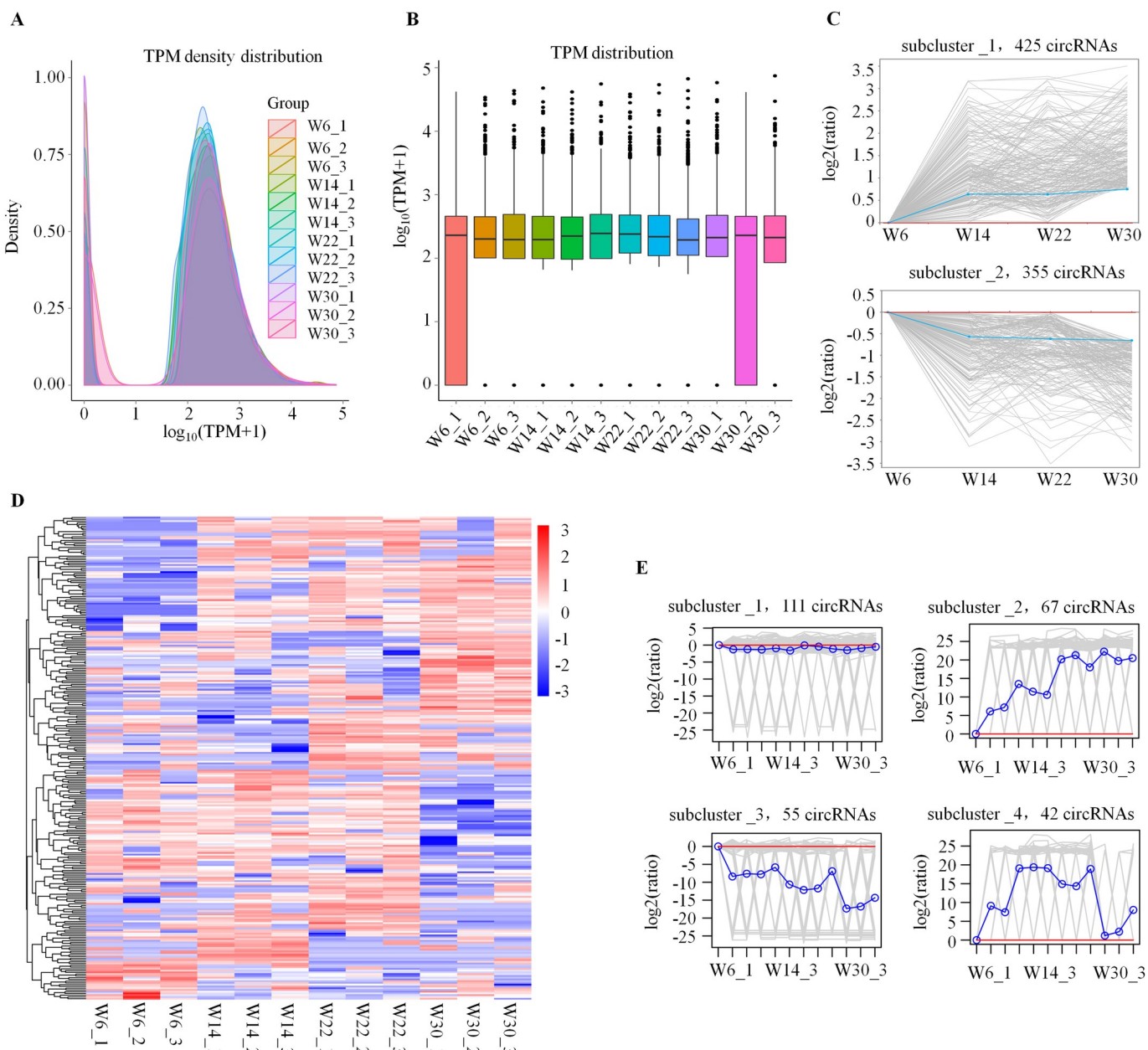

**Fig 2. Expression analysis of circRNAs during the development of abdominal fat in Gushi chickens.** (A) Density distribution of TPM values of circRNAs in each library. (B) Box plot of TPM values of circRNAs in 12 libraries. (C) Two subclusters of the circRNA expression profile in Gushi chicken abdominal adipose tissue. K-means clustering was used to represent the circRNA dynamic profile. The gray line is a line chart representing the relative expression level of a given circRNA for the four developmental stages. The blue line is a line chart representing the relative mean expression level of all circRNAs for the four developmental stages. The red line is for reference; circRNAs above the red line were upregulated, and those below the red line were downregulated. (D) Hierarchical clustering of the DE circRNAs in 12 libraries. (E) K-means cluster map of the DE circRNAs.

tissue were mainly related to the regulation of fat cell differentiation between 6 and 14 weeks of age, and they functioned primarily in biological processes, such as fatty acid homeostasis, long-chain fatty-acyl-CoA metabolic process, fat pad development, and triglyceride homeostasis, after 14 weeks of age (Fig 4A). The regulatory effects of DE circRNAs were determined to be consistent with the physiological characteristics of abdominal adipose tissue development in Gushi chickens.

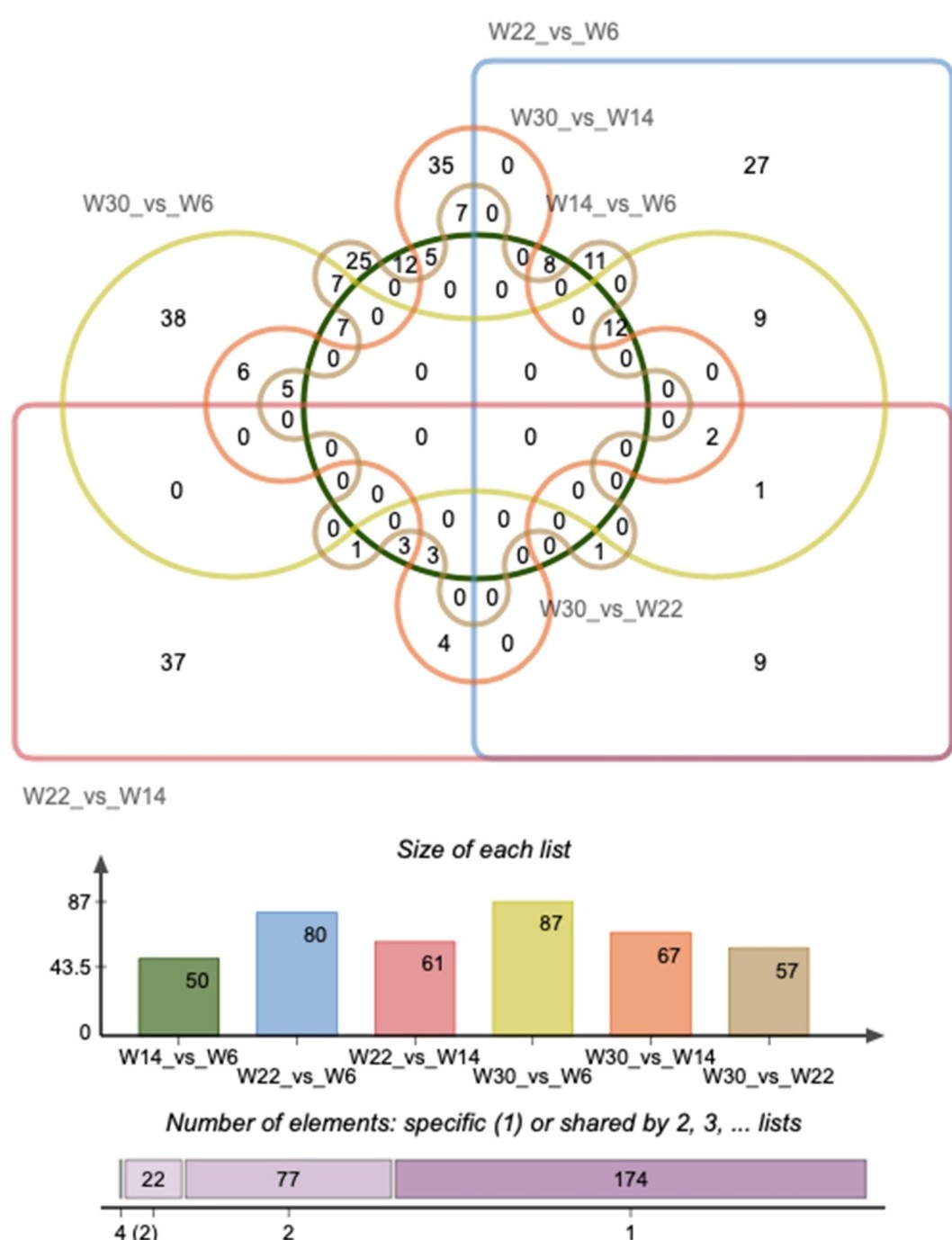

**Fig 3. Venn diagram of the DE circRNAs in six comparison groups.**

Moreover, the results of KEGG pathway analysis showed that some pathways related to lipid metabolism, such as pyruvate metabolism, fatty acid metabolism, fatty acid biosynthesis, the insulin signaling pathway and the Wnt signaling pathway, were enriched in the W22 *vs.* W6 combination (Fig 4B). The phosphatidylinositol signaling system and glycerolipid metabolism were significantly enriched in the W30 *vs.* W22 combination (Fig 4C). In particular, we

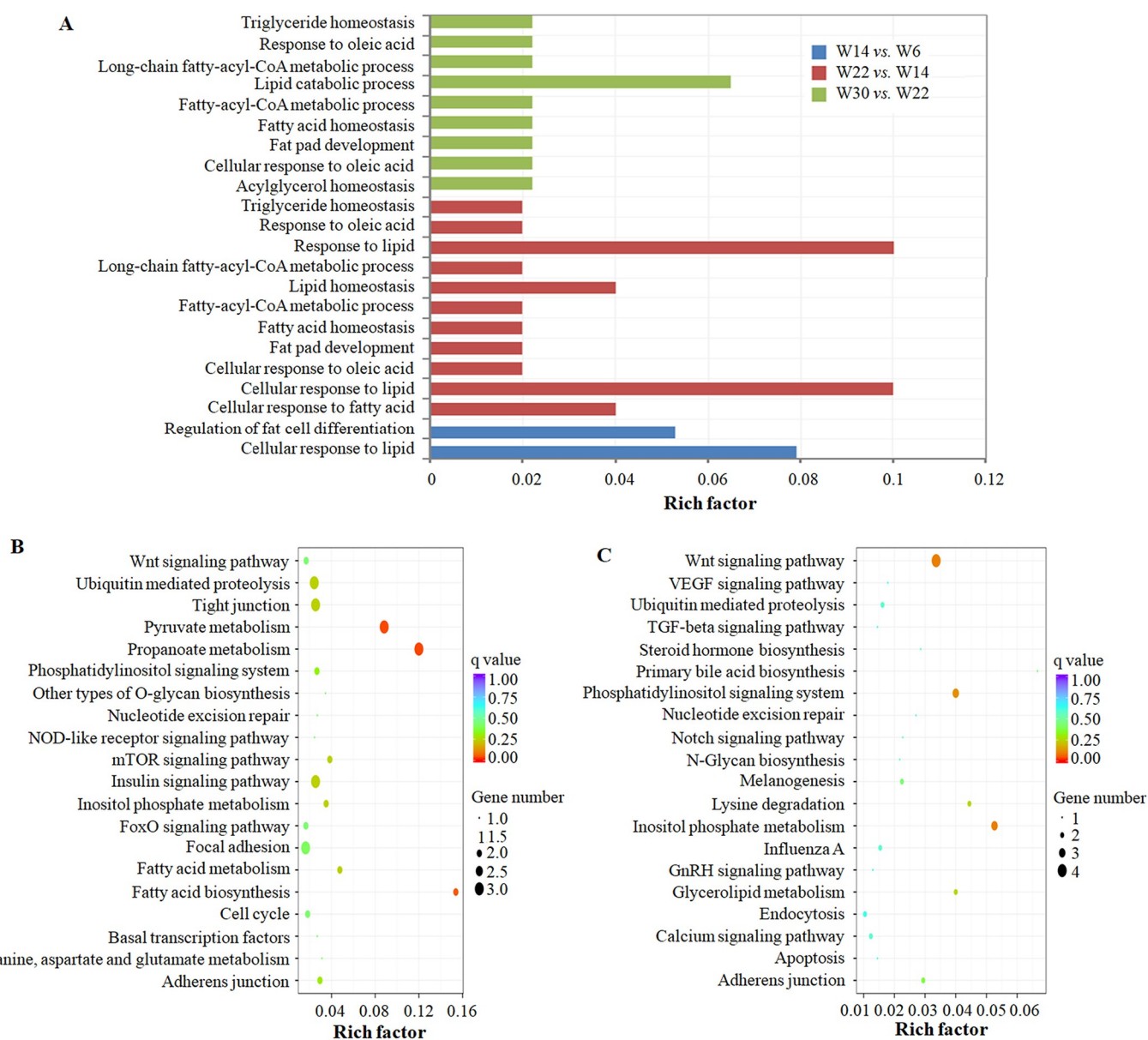

**Fig 4. Functional enrichment analysis of the parental genes of DE circRNAs.** (A) The objective was to compare the different biological processes related to lipid metabolism in W14 and W06, W22 and W14, and W30 and W22 combinations. The y-axis represents the GO terms that were significantly enriched in different comparisons, and the x-axis represents the enrichment factors in the corresponding GO terms. (B) The first 20 KEGG pathways were enriched in the W22 *vs*. W6 combination. (C) The first 20 KEGG pathways were enriched in the W30 *vs*. W22 combination.

found that the enrichment pathways propionic acid metabolism, fatty acid biosynthesis and pyruvate metabolism formed an interactive regulatory network through the DE circRNAs and their parental genes, as well as miRNAs and their targets (Fig 5). Acetyl-CoA carboxylase alpha (*ACACA*) and acetyl-CoA carboxylase beta (*ACACB*) were significantly enriched in all three pathways, and they were the parental genes for gga_circ_0001526 and gga_circ_0001148, respectively. MiR-125b-5p, miR-1559-5p, miR-1736-5p and miR-187-3p had target sites with *ACACA*. Acyl-CoA synthetase short-chain family member 3 (*ACSS3*) was the parental gene of gga_circ_0002744 and gga_circ_0002746, and miR-193a-3p, miR-193b-3p, miR-194 and miR-

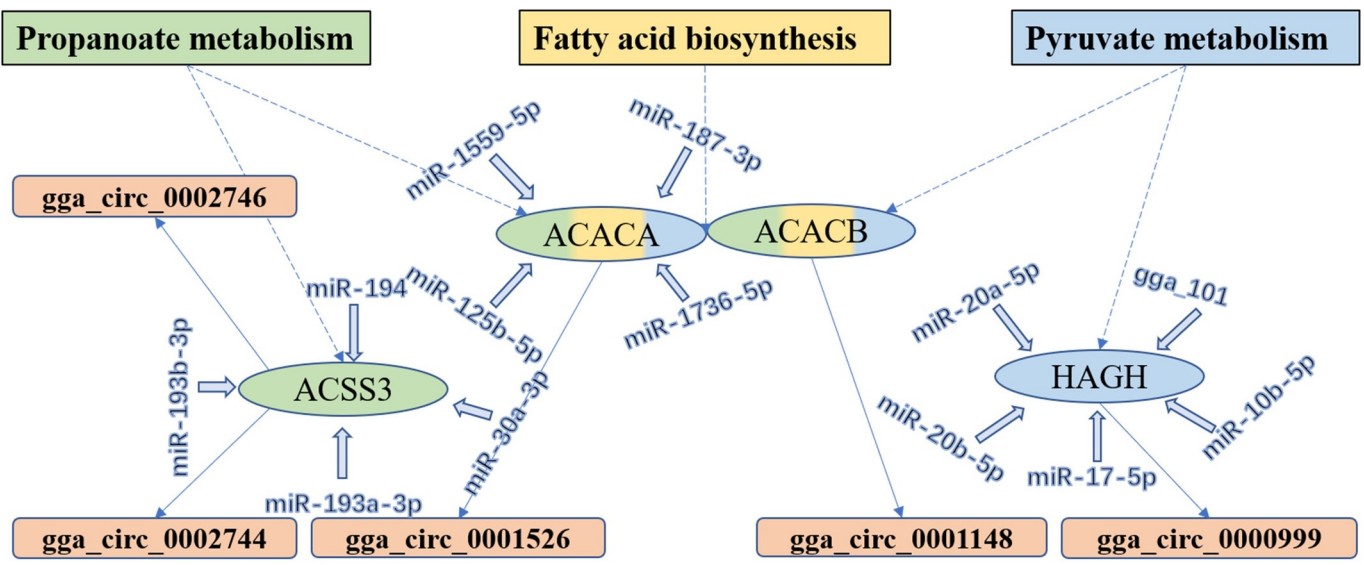

**Fig 5. Interaction network of DE mRNAs and DE circRNAs across three pathways in the process of lipid metabolism.** The genes within the propanoate metabolism, fatty acid biosynthesis, and pyruvate metabolism pathways are shown in green, yellow, and blue, respectively. The DE mRNAs are the parental gene of DE circRNAs. There are target interactions between DE mRNAs and miRNAs.

30a-3p had target sites. These results indicate that circRNAs may participate in the regulation of abdominal fat development or deposition through their parental genes and corresponding pathways in chickens.

## MiRNA target sites on circRNAs

It is well-known that the most significant function of circRNAs is to serve as miRNA sponges and regulate the expression of target genes by inhibiting miRNA activity [42]. Therefore, we predicted the binding sites of known chicken miRNAs on the identified circRNA sequences (S3 File). The results showed that there were 61,988 miRNA binding sites in 1,766 circRNAs with an average of 35 miRNA binding sites on each circRNA and a maximum of 185 miRNA binding sites. These sites could be bound by 1,235 known miRNAs. Thus, a single circRNA can bind to a considerable number of miRNAs, and a miRNA can also target many circRNAs simultaneously. These results are consistent with previous studies in human visceral preadipocytes and adipocytes [7] and yak adipocytes [11]. This finding suggests that circRNAs, as miRNA sponges, have a complex regulatory relationship in abdominal fat tissue development or adipogenesis in chickens.

Literature mining determined that 35 miRNAs of the 1,235 miRNAs with binding sites on circRNAs, such as let-7b, miR-101-3p, miR-103-3p, and miR-10b-5p, were related to lipid metabolism or adipogenesis [43–68]. These miRNAs were also expressed during the development of abdominal fat in Gushi chickens [69]. To characterize the complex interaction between circRNAs and miRNAs, we constructed a circRNA-miRNA interaction network using these 35 miRNAs (Fig 6). In the network, some circRNAs contained multiple target sites for more than four different miRNAs, such as gga_circ_0000582, gga_circ_0001623, gga_circ_0008796, and gga_circ_0008809. Among these miRNAs decoyed by circRNAs, miR-20a can regulate adipocyte differentiation by targeting lysine-specific demethylase 6b [43], miR-144-3p promotes fat formation by releasing *C/EBPα* from *Klf3* and *CtBP2* [44], and miR-148a can modulate adipocyte differentiation of mesenchymal stem cells [45]. In addition, miR-204

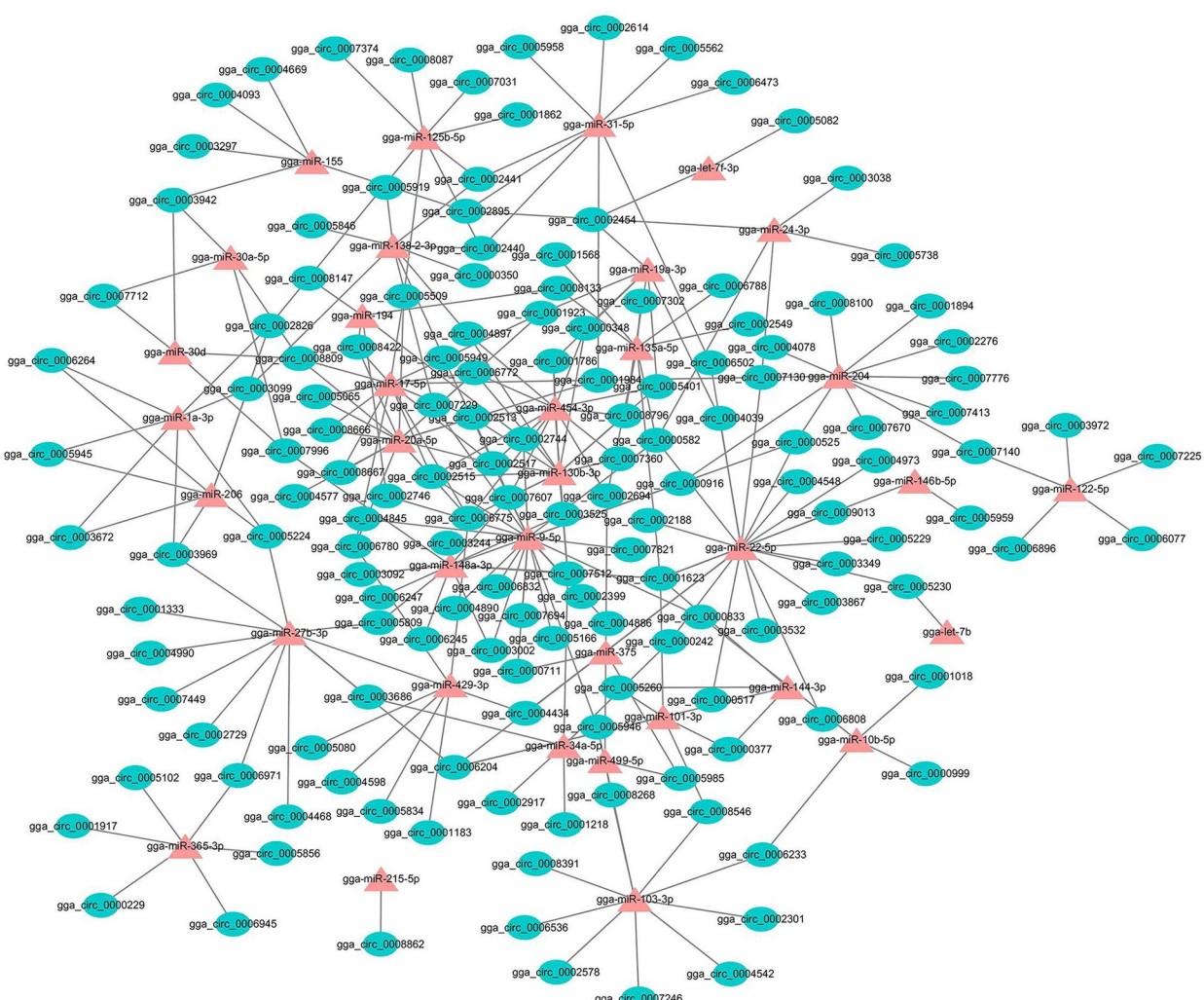

**Fig 6. CircRNA-miRNA interaction networks related to lipid metabolism or deposition.** circRNAs and miRNAs are represented by ellipses and triangles, respectively.

can target 11 different circRNAs, such as gga_circ_0007413, gga_circ_0007670, gga_-circ_0007776, and gga_circ_0008100, in the network, and it has been shown to be involved in the regulation of bovine adipocyte proliferation, differentiation, and apoptosis [46]. These studies indicate that circRNAs may play endogenous competitive regulatory roles through interaction with multiple miRNAs during the development of chicken abdominal fat tissue.

## IRES elements on circRNAs

The internal ribosome entry site (IRES) is an important regulatory element for RNA translation without relying on the 5′ cap structure, and it can mediate the assembly of ribosomal larger and smaller subunits. Studies have shown that circRNAs contain IRES, and some circRNAs can also encode proteins or functional peptides *in vivo* and *in vitro* [70–72]. Therefore, we predicted the potential IRES elements in the identified circRNA sequences. There were IRES elements on 1,108 circRNAs, accounting for 62.71% of the predicted circRNA number (Fig 7). The top 10 circRNAs in the predicted scores were gga_circ_0007953, gga_circ_0005736, gga_circ_0005910, gga_circ_0001863, gga_circ_0001990, gga_circ_0005598, gga_

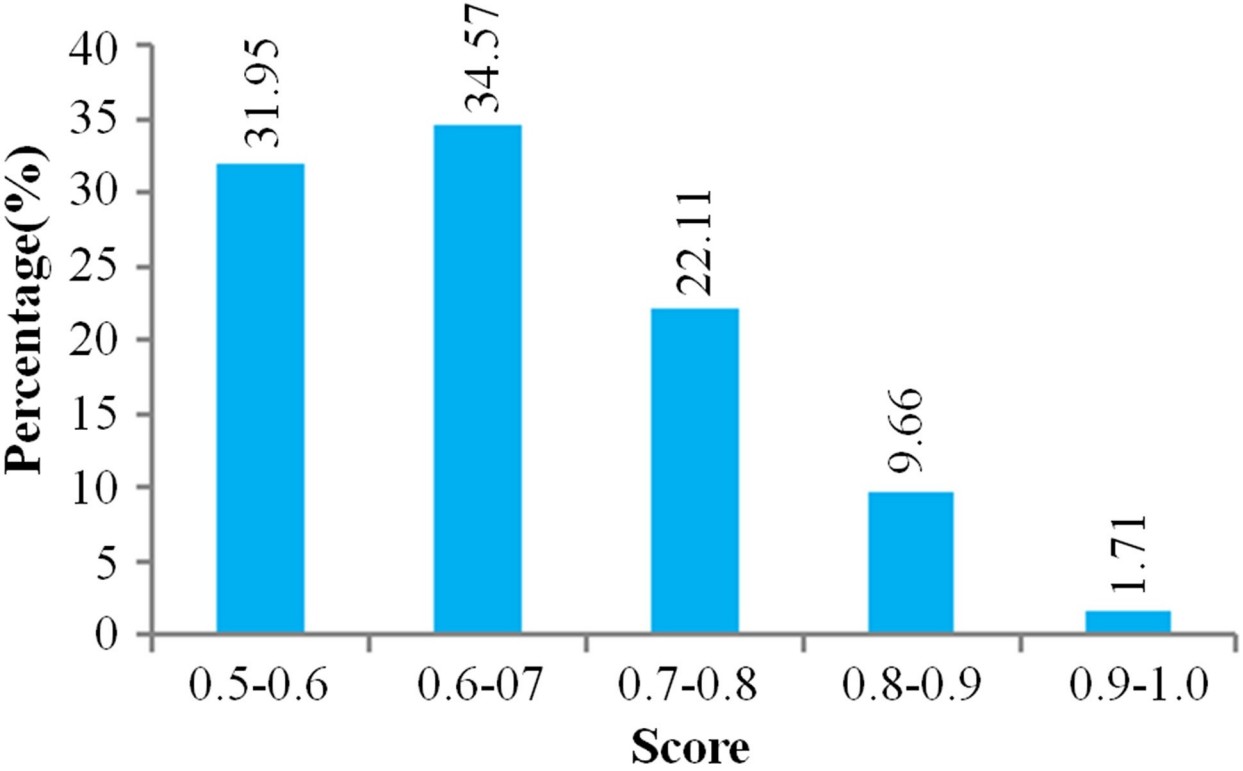

**Fig 7. Predicted score distribution of circRNAs with the IRES element.** The x-axis indicates the interval of the predicted score, and the y-axis represents the percentage of circRNAs within the corresponding predicted score interval for all circRNAs with IRES. CircRNAs with predicted scores of > 0.5 were considered to have IRES elements. A total of 1,108 circRNAs with IRES were identified.

circ_0006417, gga_circ_0007094, gga_circ_0004846, and gga_circ_0002156. These results indicated that most circRNAs in chicken abdominal adipose tissue may have the potential to encode peptides or proteins.

## CircRNA-miRNA-gene networks during abdominal adipose development

Abdominal fat tissue development is a complex process participating in many biological processes and cellular events, which are regulated by multiple signaling pathways. It is known that circRNAs regulate the function of miRNAs by a ceRNA mechanism [73], thereby impacting the posttranscriptional regulation of miRNAs in downstream genes [74] and regulating biological processes by affecting signaling pathways [4]. For this reason, we analyzed the interactions among genes, miRNAs, circRNAs, and pathways during the development of Gushi chicken abdominal fat. Based on the results of KEGG enrichment analyses using the whole transcriptome sequencing data of Gushi chicken abdominal fat at 6, 14, 22, and 30 weeks of age, we identified 53, 83, and 56 circRNA-miRNA-gene pairs associated with lipid metabolism, cell proliferation and differentiation, and cell junction pathways, respectively, and constructed three ceRNA networks (Fig 8, S4 and S5 Figs). The lipid metabolism-related ceRNA network contains 32 circRNAs, 19 miRNAs, and 13 genes and involves 13 pathways, including glycerolipid metabolism, glycerophospholipid metabolism, fatty acid elongation, sphingolipid metabolism, fatty acid degradation, alpha-linolenic acid metabolism, adipocytokine signaling pathway, fatty acid metabolism, the PPAR signaling pathway, fatty acid biosynthesis, biosynthesis of unsaturated fatty acids, ether lipid metabolism and insulin signaling pathway (Fig 8).

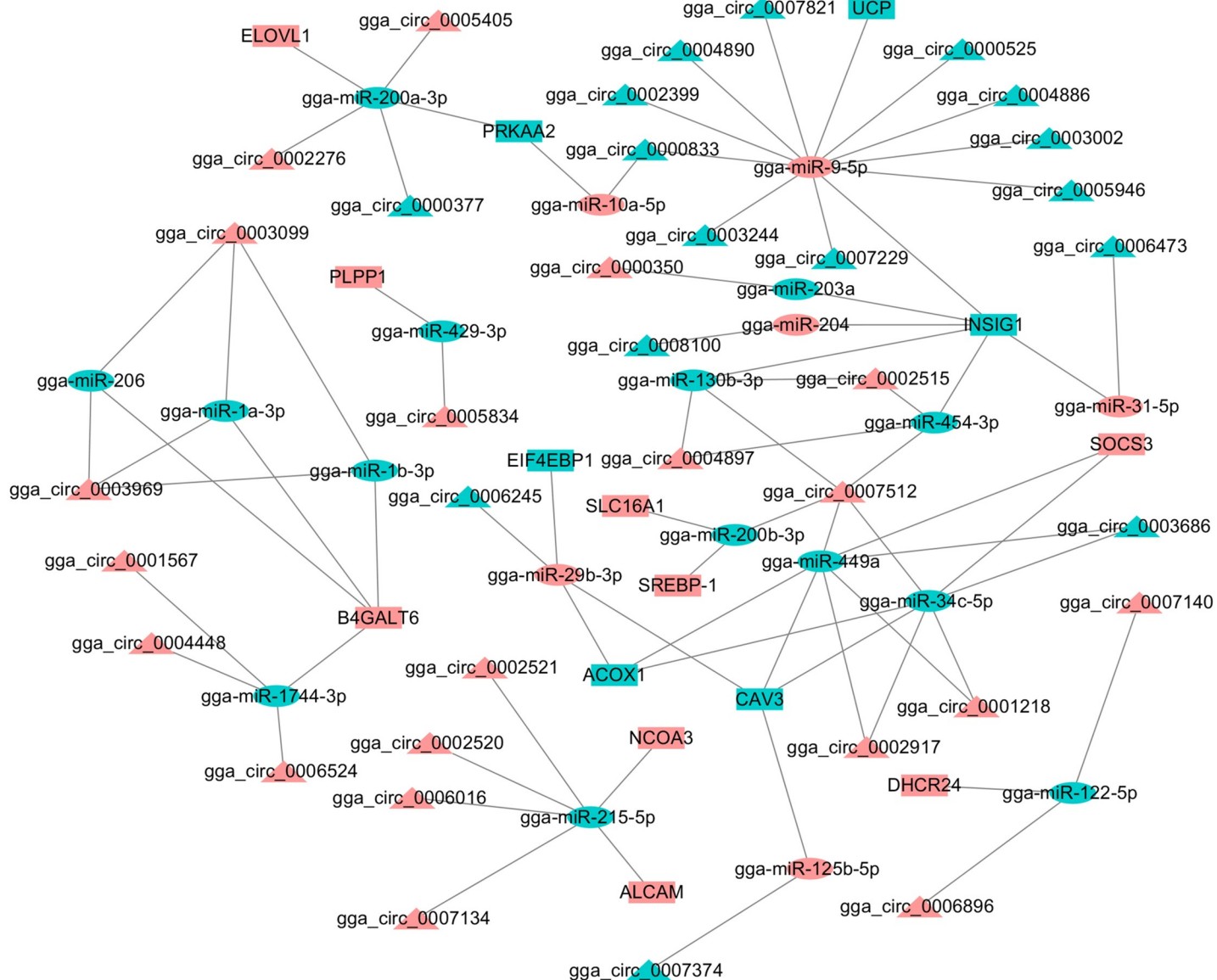

**Fig 8. CircRNA-miRNA-gene interaction network contained thirteen lipid metabolism- or deposition-related pathways.** The ellipse, triangle and box nodes represent miRNAs, circRNAs, mRNAs, respectively. Pink indicates upregulation, and blue indicates downregulation.

In these networks, some miRNAs, such as gga-miR-130b-3p, gga-miR-1744-3p, gga-miR-200a-3p, gga-miR-1a-3p, gga-miR-34c-5p, gga-miR-449a, gga-miR-454-3p, and gga-miR-9-5p, targeted multiple circRNAs, and some genes acted as components of multiple pathways and were targeted by multiple miRNAs. Meanwhile, some circRNAs, such as gga_circ_0000348, gga_circ_0000525, gga_circ_0007512, and gga_circ_0008796, interacted with more than two miRNAs. As key nodes, these factors constituted a complex ceRNA network and synergistically regulated lipid metabolism and multiple cellular events during the development of abdominal fat tissue in Gushi chickens. In addition, the ceRNA gga_circ_0002520-miR-215-5p-*NCOA3* in the lipid metabolism-related circRNA-miRNA-gene interaction network was selected and verified by luciferase reporter assay. The results showed that there were target sites of miR-215-5p in the gga_circ_0002520 sequence and *NCOA3* mRNA 3′-UTR (Fig 9),

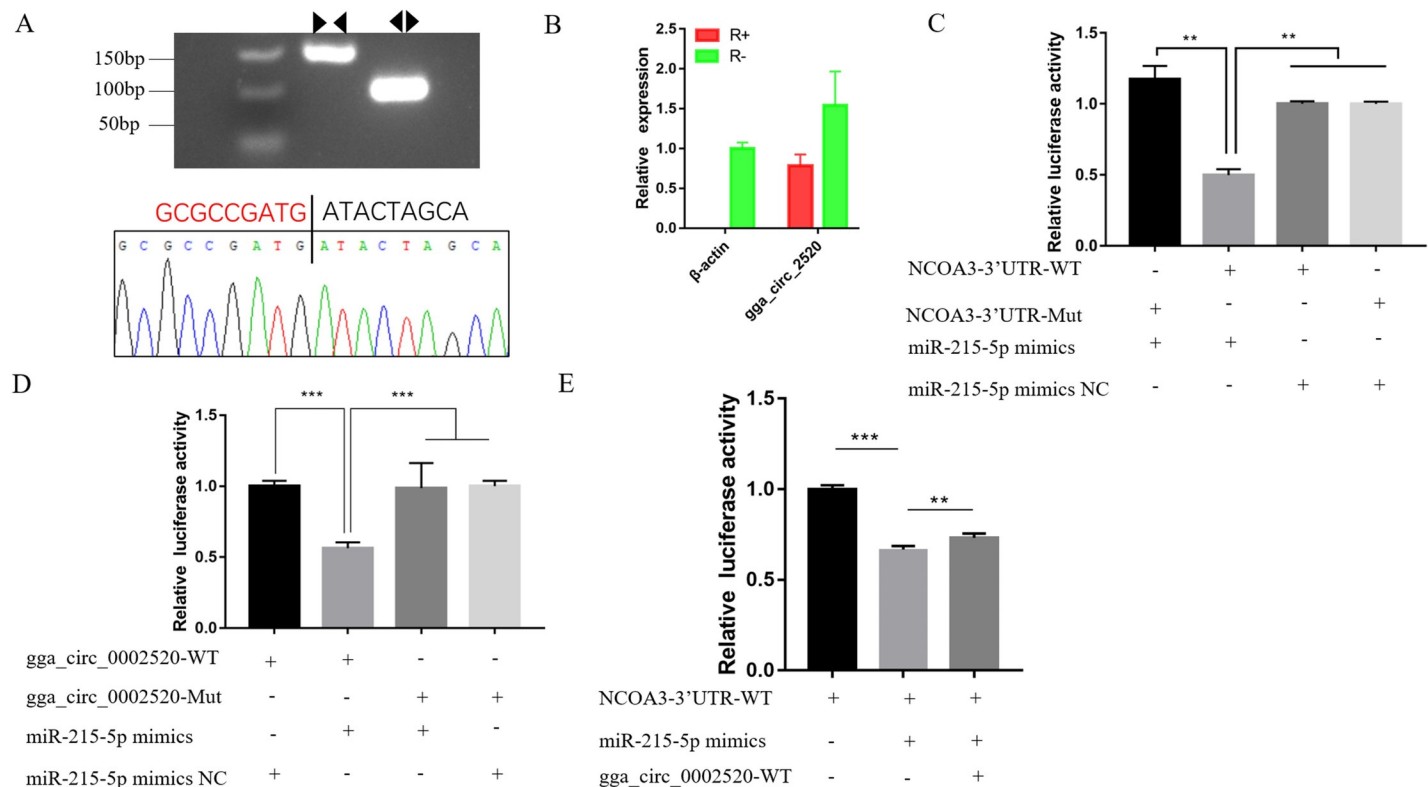

**Fig 9. Experimental verification of the ceRNA gga_circ_0002520-miR-215-5p-NCOA3.** (A) Convergent and divergent primers amplify circRNAs in cDNA. The head-to-tail junctions of gga_circ_0002520 were confirmed by Sanger sequencing. The black line indicates the position of reverse splicing. (B) qRT-PCR was used to detect the resistance of gga_circ_0002520 to RNase R digestion. *β-actin* was used as a linearity control. Data are expressed as means ± SEM (n = 3). (C) Luminescence was measured after cotransfecting the wild-type or mutant sequence of NCOA3-3′UTR with miR-215-5p mimics (or mimics NC) in DF-1 cells. (D) Luminescence was measured after cotransfecting the wild-type or mutant linear sequence of gga_circ_0002520 with miR-215-5p mimics (or mimics NC) in DF-1 cells. (E) A double luciferase assay was performed in DF-1 cells 48 h after miR-215-5p mimic transfection with NCOA3-3′UTR-WT and gga_circ_0002520-WT.

which demonstrated that the results of our circRNA-miRNA-gene interaction analysis were reliable. These results indicated that circRNAs play critical roles in the regulation of chicken abdominal fat tissue development or fat deposition through the complex networks formed by circRNAs, miRNAs and genes and corresponding pathways.

## Conclusions

The present study identified many circRNAs and DE circRNAs related to abdominal adipose tissue development and assessed the characteristics of circRNAs and their expression profiles in abdominal adipose tissue during late development in Gushi chickens. We found that circRNAs had specific expression patterns and obvious temporal characteristics in chicken abdominal adipose tissue, and most circRNAs had the potential to encode proteins or functional peptides. In addition, we also found that the parental genes of DE circRNAs were primarily enriched in some pathways related to lipid metabolism, such as pyruvate metabolism, fatty acid metabolism, and fatty acid biosynthesis, and these circRNAs play critical roles in the regulation of chicken abdominal adipose tissue development by complex ceRNA networks. Our results provide novel insight and a valuable resource helping to elucidate the regulatory roles played by circRNAs in chicken abdominal adipose tissue. Further research is required to clarify the function of the identified circRNAs in chicken abdominal adipose tissue development or deposition.

## Supporting information

**S1 Fig. Quality evaluation and processing of sequencing data.** (A) The sequencing error rate of each base position. (B) The content of each base during the sequencing process. (C) The proportion of different types of raw data. (D) Classification of mapped reads in 12 libraries.
(TIF)

**S2 Fig. Characteristics of the identified circRNAs.** (A) Length distribution. (B) The distribution of circRNAs in different chromosomes.
(TIF)

**S3 Fig. Dynamic expression profile of the identified circRNAs.**
(TIF)

**S4 Fig. CircRNA-miRNA-gene interaction network contains five pathways related to cell proliferation and differentiation.** The ellipse, triangle and box nodes represent DE miRNAs, DE circRNAs, and DE mRNAs, respectively. Pink indicates upregulation, and blue indicates downregulation. The five pathways are the Wnt signaling pathway, FoxO signaling pathway, p53 signaling pathway, TGF-beta signaling pathway and MAPK signaling pathway.
(TIF)

**S5 Fig. CircRNA-miRNA-gene interaction network containing five cell junction-related pathways.** The ellipse, triangle and box nodes represent miRNAs, circRNAs, mRNAs, respectively. Pink indicates upregulation, and blue indicates downregulation. The five pathways are cell adhesion molecules, focal adhesion, adherens junctions, tight junctions and gap junctions.
(TIF)

**S1 Table. Primers used for the validation of circRNAs.**
(DOCX)

**S1 File. Details of the circRNAs identified in this study.**
(XLSX)

**S2 File. Details of the significant differentially expressed circRNAs identified in this study.**
(XLSX)

**S3 File. Binding site analysis of miRNAs on the circRNAs.**
(XLSX)

**S1 Raw images.**
(PDF)

## Author Contributions

**Conceptualization:** Yinli Zhao.

**Data curation:** Yuanfang Li.

**Investigation:** Guirong Sun.

**Methodology:** Bin Zhai, Ruirui Jiang.

**Software:** Bin Zhai, Yadong Tian.

**Supervision:** Xiangtao Kang.

**Validation:** Shengxin Fan, Pengtao Yuan, Yanbin Wang.

**Visualization:** Xiaojun Liu.

**Writing – original draft:** Wenjiao Jin.

**Writing – review & editing:** Guoxi Li.

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
