## [Decision Letter · Decision Letter 0]

4 Feb 2021

PONE-D-20-33474

Characterization and regulatory networks of circRNAs in developing abdominal adipose tissues in Chinese Gushi chicken

PLOS ONE

Dear Dr. Li,

Thank you for submitting your manuscript to PLOS ONE. After careful consideration, we feel that it has merit but does not fully meet PLOS ONE’s publication criteria as it currently stands. Therefore, we invite you to submit a revised version of the manuscript that addresses the points raised during the review process.

Both reviewers have a large number of editorial comments that need to be addressed in your revision.  I suggest that you ask a primary English speaker to proof read the revision before submission.  Further, there are comments about the title  not quite matching the narrative of your manuscript.  Review 1 also has suggested limiting the number of tables/figures and adding the rest as  an Appendix as well as limiting your Results and Discussion to the data presented.  We ask that in your revision cover letter that you provide a rebuttal or answer to each comment/question  made by the reviewers.   

We look forward to receiving your revised manuscript.

Kind regards,

Michael H. Kogut, Ph.D.

Academic Editor

PLOS ONE

Journal Requirements:

2)  PLOS ONE now requires that authors provide the original uncropped and unadjusted images underlying all blot or gel results reported in a submission’s figures or Supporting Information files. This policy and the journal’s other requirements for blot/gel reporting and figure preparation are described in detail at https://journals.plos.org/plosone/s/figures#loc-blot-and-gel-reporting-requirements and https://journals.plos.org/plosone/s/figures#loc-preparing-figures-from-image-files. When you submit your revised manuscript, please ensure that your figures adhere fully to these guidelines and provide the original underlying images for all blot or gel data reported in your submission. See the following link for instructions on providing the original image data: https://journals.plos.org/plosone/s/figures#loc-original-images-for-blots-and-gels.

3) PLOS requires an ORCID iD for the corresponding author in Editorial Manager on papers submitted after December 6th, 2016. Please ensure that you have an ORCID iD and that it is validated in Editorial Manager. To do this, go to ‘Update my Information’ (in the upper left-hand corner of the main menu), and click on the Fetch/Validate link next to the ORCID field. This will take you to the ORCID site and allow you to create a new iD or authenticate a pre-existing iD in Editorial Manager. Please see the following video for instructions on linking an ORCID iD to your Editorial Manager account: https://www.youtube.com/watch?v=_xcclfuvtxQ

Reviewers' comments:

Reviewer's Responses to Questions

**Comments to the Author**

1. Is the manuscript technically sound, and do the data support the conclusions?

Reviewer #1: Yes

Reviewer #2: Yes

2. Has the statistical analysis been performed appropriately and rigorously? 

Reviewer #1: No

Reviewer #2: Yes

3. Have the authors made all data underlying the findings in their manuscript fully available?

Reviewer #1: Yes

Reviewer #2: Yes

4. Is the manuscript presented in an intelligible fashion and written in standard English?

Reviewer #1: No

Reviewer #2: Yes

5. Review Comments to the Author

Reviewer #1: The authors investigate different time points’ abdominal adipose tissues of old Gushi chicken by RNA-seq. They identified 275 circRNAs with significant differentially expressed（DE）were identified from 6 combinations. GO term and KEGG pathways enrichment analysis showed DE circRNAs related to lipid metabolism or deposition. The manuscript should be of interest to the readers. However, the results are not well presented and the statistical analysis would need to be revised. Some issues need to be addressed:

1. The title is not properly summary the manuscript.

2. There are too many figures in the manuscript which can’t point out the key points or the novel discovery at all. The authors have to simplify the manuscript. Like the part “Library sequencing and quality evaluation” could be move to method, or describe it briefly. The fig for this part should be go to supplementary fig or probably not needed.

3. The authors only mentioned the total numbers of circRNAs they identified, can you list the identified numbers of each time point. The authors also should perform analysis for the variations of the different time points.

4. The authors should have a figure panel to show the dynamic circRNA profile of abdominal adipose tissue.

5. For the circRNA-miRNA-gene network, can you choose one circRNA-miRNA-gene pathway to validate your data?

6. Did the authors identified some new circRNAs? If yes, they should demonstrate them and discuss them in the discussion part.

Other Comments

1. Line 246， 30 vs 22, should W30 vs W22.

2. The figure quality was very low. Some pictures are pretty blurred.

Reviewer #2: In this manuscript, the circRNAs expression profile characteristics of Gushi chicken abdominal adipose tissues was described where many circRNAs involved in chicken adipose tissue development or fat deposition were obtained, and ceRNA regulatory networks in developing abdominal adipose tissues were constructed. The research is very interesting and innovative. However, there are still existing some problems:

(1) Reference list: Information of some references are incomplete and should be supplemented. Please make a complete revision of the reference list according to the magazine format requirements.

(2) Check the whole manuscript to make sure the gene name is italicized. For example, line 481, 514, 569-570, and etc..

(3) Version information of all the softwares used in the study should be provided.

(4) In line 87, the source information of the "Epicentre Ribo-ZerotM kit" used in the study were incomplete. Please supplement and check the whole manuscript.

(5) Check out the whole manuscript for the consistant usage of P value or p value. For example, the letter "P" in line 115 is lowercase; the "P" in line 121 is capitalized.

(6) The names of "miRanda software" in line 123 and line 131 should be written in a consistent way.

(7) The first letter of the word in the sentence is lowercase. Please check the whole manuscript and correct it. For example, the "Count" in line 112, the "Proportion" in line 175, and etc..

(8) Grammar errors should be avoided. For example, a phrase " after 14 weeks of age" should be deleted from the sentence "After 14 weeks of age, the biological functions of DE circRNAs mainly participated in biological processes such as fatty acid homeostasis, long-chain fatty-acyl-CoA metabolic process, fat pad development, and triglyceride homeostasis after 14 weeks of age" in line 282-285. Please check the whole manuscript.

(9) The spelling mistakes of the word in the sentence should be avoided. For example, the " GO items " in line 565, the "lipogene-sis" in abstract, and etc.. Please check the whole manuscript.

6. PLOS authors have the option to publish the peer review history of their article (what does this mean?). If published, this will include your full peer review and any attached files.

Reviewer #1: **Yes: **Xing Liu

Reviewer #2: No

---

## [Author Response · Author response to Decision Letter 0]

12 Mar 2021

Dear editor and reviewers,

Thank you very much for your hard work on my manuscript. According to the review comments of editors and reviewers and the requirements of the magazine, we have made a thorough revision of the manuscript entitled “Characterization and regulatory networks of circRNAs in developing abdominal adipose tissues in Chinese Gushi chicken” (PONE-D-20-33474). The contents are as follows：

1. According to the review comments of reviewer 1, we changed the title to " Characteristics and expression profiles of circRNAs during abdominal adipose tissue development in Chinese Gushi chickens".

2. The abstract and conclusion have been rewritten to simplify the content.

3. In order to simplify the content of the manuscript, the results and discussion were merged into "Results and Discussion", and the integrated content has been rewritten.

4. Modified and improved some secondary titles.

5. According to the comments of reviewer 1, the number of pictures in the manuscript were reduced and the figures 1, 2, 11 and 12 were taken as additional files.

6. According to the comments of reviewer 1, the verification result of a circRNA-miRNA-gene pathway was added, and figure 9 was added to show the result.

7. The reference list has been adjusted and improved.

8. The additional files have been improved .

9. The language of revised manuscript has been refined and polished by the Novo Biology.

In addition, the sections marked in red are the modified contents in the “Revised Manuscript with Track Changes”.

The response to comments from Reviewer 1

Thank you for your objective evaluation of our manuscript. We have revised the manuscript in accordance with your comments.

Point 1: The title is not properly summary the manuscript.

Response 1: Thank you for your comments. We have modified the title to " Characteristics and expression profiles of circRNAs during abdominal adipose tissue development in Chinese Gushi chickens".

Point 2: There are too many figures in the manuscript which can’t point out the key points or the novel discovery at all. The authors have to simplify the manuscript. Like the part “Library sequencing and quality evaluation” could be move to method, or describe it briefly. The fig for this part should be go to supplementary fig or probably not needed.

Response 2: Thank you so much for your comments. We have revised the original manuscript thoroughly according to your opinion. The results and discussion in the original manuscript are merged into "results and discussion", and the integrated content has been rewritten. The redundant languages have been removed. At the same time, the number of figures in the manuscript have been reduced, and figures 1, 2, 11 and 12 in the original manuscript have been taken as supplementary information. See the revised manuscript for details.

Point 3: The authors only mentioned the total numbers of circRNAs they identified, can you list the identified numbers of each time point. The authors also should perform analysis for the variations of the different time points.

Response 3: We apologize for not describing this clearly. We have supplemented the number of circRNAs identified at each time point (Page 6, Lines 208-210), and at the same time, we have also analyzed the variations at different time points (Page 7, Lines 263-267). The contents are as follows：

In total, 1,637, 1,711, 1,735 and 1,663 circRNAs were expressed in the abdominal adipose tissue of Gushi chicken at 6, 14, 22, and 30 weeks of age, respectively. (Page 6, Lines 208-210)

Meanwhile, we found that 1,518 of the identified circRNAs were expressed in all four developmental stages. The expression profiles of these circRNAs exhibited clear temporal characteristics and were divided into eight different subclusters (Fig. 2C and Fig. S3). We also found that one, three, six, and two circRNAs were expressed specifically in abdominal adipose tissue at 6, 14, 22, and 30 weeks of age, respectively (File S1). (Page 7, Lines 263-267)

Point 4: The authors should have a figure panel to show the dynamic circRNA profile of abdominal adipose tissue.

Response 4: Thank you for your comments. We have analyzed the dynamic expression profile of circRNA in abdominal adipose tissue (Page 7, Lines 263-265 and Fig 2C , S3 Fig). The contents are as follows：

Meanwhile, we found that 1,518 of the identified circRNAs were expressed in all four developmental stages. The expression profiles of these circRNAs exhibited clear temporal characteristics and were divided into eight different subclusters (Fig. 2C and Fig. S3). (Page 7, Lines 263-265)

Point 5: For the circRNA-miRNA-gene network, can you choose one circRNA-miRNA-gene pathway to validate your data?

Response 5: Thank you for your valuable comments. A ceRNA network, gga_circ_0002520-miR-215-5p-NCOA3, has been verified (Page 11, Lines 422-427 and Fig 9). The contents are as follows：

In addition, the ceRNA gga_circ_0002520-miR-215-5p-NCOA3 in the lipid metabolism-related circRNA-miRNA-gene interaction network was selected and verified by luciferase reporter assay. The results showed that there were target sites of miR-215-5p in the gga_circ_0002520 sequence and NCOA3 mRNA 3´-UTR (Fig. 9), which demonstrated that the results of our circRNA-miRNA-gene interaction analysis were reliable. (Page 11, Lines 422-427)

Point 6: Did the authors identified some new circRNAs? If yes, they should demonstrate them and discuss them in the discussion part.

Response 6: The circRNAs identified in this study are all new circRNAs. In the manuscript, we selected 10 newly identified circRNAs for verification and these results are shown in "Results and Discussion", and we have discussed these contents. For details, see line 207-208, lines 218-228, Fig 1 and Fig 9A and B of the revised manuscript. The contents are as follows：

A total of 1,766 novel circRNAs were identified from the 12 cDNA libraries in Gushi chicken abdominal adipose tissue (File S1). (Page 6, Lines 207-208)

Previously, several studies have reported circRNAs expressed in adipose tissue, preadipocytes and adipocytes in different anatomical parts of humans and several other species, such as pigs, yaks, and buffalo [7-12]. These studies indicate that a large quantity of circRNAs participate in the regulation of fat tissue development or adipogenesis. The results of this study are consistent with the findings of previous studies, suggesting that the development or deposition of chicken abdominal fat is also regulated by many circRNAs. Ten circRNAs were randomly selected and verified by reverse transcription PCR analysis, Sanger sequencing and RNase R digestion tests. These results proved that the identification of circRNAs in this study was accurate and reliable (Fig. 1 and Fig. 9, A and B). Therefore, the results of this study provide a basis for further research on the molecular mechanism for the regulation of chicken abdominal adipogenesis by circRNAs. (Page 6-7, Lines 218-228)

Point 7: Line 246， 30 vs 22, should W30 vs W22.

Response 7: Thank you for your comments. It has been revised, see lines 269 of the revised manuscript.

Point 8: The figure quality was very low. Some pictures are pretty blurred.

Response 8: Thank you for your comments. We have improved the quality of the pictures in accordance with the requirements of the magazine format.

The response to comments from Reviewer 2

Thank you for your objective evaluation of our manuscript. We have revised the manuscript in accordance with your comments.

Point 1: Reference list: Information of some references are incomplete and should be supplemented. Please make a complete revision of the reference list according to the magazine format requirements.

Response 1: Thanks for your comments. We have supplemented and improved the reference list based on your comments and journal format requirements. See the revised reference list No. 4, 8-9, 11-12, 16, 18-25, 28, 32, 36-38, 40-41, 43-48, 50, 58-60, 66, 70.

Point 2: Check the whole manuscript to make sure the gene name is italicized. For example, line 481, 514, 569-570, and etc. 

Response 2: Thank you for your comments. Based on your opinion, we have done a comprehensive inspection and modification. See lines 289 and 360 of the revised manuscript.

Point 3: Version information of all the softwares used in the study should be provided.

Response 3: Based on your opinion, we have improved the version information of the software. See lines 101, 104, 118, 123, 127, 129, 135, 143, and 154 of the revised manuscript.

Point 4: In line 87, the source information of the "Epicentre Ribo-ZerotM kit" used in the study were incomplete. Please supplement and check the whole manuscript.

Response 4: Thank you for your comments. According to your opinion, we have supplemented the source information of the reagents used. Please refer to line 89 and 93 of the revised manuscript.

Point 5: Check out the whole manuscript for the consistant usage of P value or p value. For example, the letter "P" in line 115 is lowercase; the "P" in line 121 is capitalized.

Response 5: Thank you for your comments. We have unified the "P" to lowercase. See lines 119 and 125 of the revised manuscript.

Point 6: The names of "miRanda software" in line 123 and line 131 should be written in a consistent way.

Response 6: Thank you for your comments. We have corrected them. See lines 127 and 135 of the revised manuscript.

Point 7: The first letter of the word in the sentence is lowercase. Please check the whole manuscript and correct it. For example, the "Count" in line 112, the "Proportion" in line 175, and etc..

Response 7: Thank you for your comments. We have thoroughly checked and revised the manuscript, and corrected the problems you mentioned.

Point 8: Grammar errors should be avoided. For example, a phrase " after 14 weeks of age" should be deleted from the sentence "After 14 weeks of age, the biological functions of DE circRNAs mainly participated in biological processes such as fatty acid homeostasis, long-chain fatty-acyl-CoA metabolic process, fat pad development, and triglyceride homeostasis after 14 weeks of age" in line 282-285. Please check the whole manuscript.

Response 8: Thank you for your comments. We have corrected the error you mentioned, see lines 300-303 of the revised manuscript. At the same time, the revised manuscript was sent to a professional company for language editing to avoid grammatical errors.

Point 9: The spelling mistakes of the word in the sentence should be avoided. For example, the " GO items " in line 565, the "lipogene-sis" in abstract, and etc.. Please check the whole manuscript.

Response 9: Thank you for your comments. We have checked and corrected the spelling of the words in the manuscript. Meanwhile, the revised manuscript was sent to a professional company for language editing to avoid spelling errors.

---

## [Editor Report · Decision Letter 1]

16 Mar 2021

Characteristics and expression profiles of circRNAs during abdominal adipose tissue development in Chinese Gushi chickens

PONE-D-20-33474R1

Dear Dr. Li,

We’re pleased to inform you that your manuscript has been judged scientifically suitable for publication and will be formally accepted for publication once it meets all outstanding technical requirements.

Kind regards,

Michael H. Kogut, Ph.D.

Academic Editor

PLOS ONE
---

## [Editor Report · Acceptance letter]

6 Apr 2021

PONE-D-20-33474R1 

Characteristics and expression profiles of circRNAs during abdominal adipose tissue development in Chinese Gushi chickens 

Dear Dr. Li:

I'm pleased to inform you that your manuscript has been deemed suitable for publication in PLOS ONE. Congratulations! Your manuscript is now with our production department. 

Kind regards, 

on behalf of

Dr. Michael H. Kogut 

Academic Editor

PLOS ONE